# Adsorptive Membrane for Boron Removal: Challenges and Future Prospects

**DOI:** 10.3390/membranes12080798

**Published:** 2022-08-18

**Authors:** Shaymala Mehanathan, Juhana Jaafar, Atikah Mohd Nasir, Roshanida A. Rahman, Ahmad Fauzi Ismail, Rosli Md Illias, Mohd Hafiz Dzarfan Othman, Mukhlis A Rahman, Muhammad Roil Bilad, Muhammad Nihal Naseer

**Affiliations:** 1Advanced Membrane Technology Research Center (AMTEC), Faculty of Engineering, School of Chemical and Energy Engineering, Universiti Teknologi Malaysia, Johor Bahru 81310, Malaysia; 2Center for Diagnostic, Therapeutic and Investigative Studies (CODTIS), Faculty of Health Sciences, Universiti Kebangsaan Malaysia, Bangi 43600, Malaysia; 3Institute of Bioproduct Development (IBD), Universiti Teknologi Malaysia, Johor Bahru 81310, Malaysia; 4Faculty of Integrated Technologies, Universiti Brunei Darussalam, Gadong BE1410, Brunei; 5Department of Engineering Sciences, National University of Sciences and Technology (NUST), Islamabad 44000, Pakistan

**Keywords:** adsorptive membrane, boron, dual-layered membrane, surface modification, water treatment

## Abstract

The complexity of removing boron compounds from aqueous systems has received serious attention among researchers and inventors in the water treating industry. This is due to the higher level of boron in the aquatic ecosystem, which is caused by the geochemical background and anthropogenic factors. The gradual increase in the distribution of boron for years can become extremely toxic to humans, terrestrial organisms and aquatic organisms. Numerous methods of removing boron that have been executed so far can be classified under batch adsorption, membrane-based processes and hybrid techniques. Conventional water treatments such as coagulation, sedimentation and filtration do not significantly remove boron, and special methods would have to be installed in order to remove boron from water resources. The blockage of membrane pores by pollutants in the available membrane technologies not only decreases their performance but can make the membranes prone to fouling. Therefore, the surface-modifying flexibility in adsorptive membranes can serve as an advantage to remove boron from water resources efficiently. These membranes are attractive because of the dual advantage of adsorption/filtration mechanisms. Hence, this review is devoted to discussing the capabilities of an adsorptive membrane in removing boron. This study will mainly highlight the issues of commercially available adsorptive membranes and the drawbacks of adsorbents incorporated in single-layered adsorptive membranes. The idea of layering adsorbents to form a highly adsorptive dual-layered membrane for boron removal will be proposed. The future prospects of boron removal in terms of the progress and utilization of adsorptive membranes along with recommendations for improving the techniques will also be discussed further.

## 1. Introduction

The collections of genuine treatments related to water and wastewater systems impact and influence both environmental hygiene and the global economy. The increase in the global population has increased the consumption of water. Additionally, the agricultural growth and industrialization that have been happening for years cause the quality and purity of the water to deteriorate [1]. This is due to the existence of salinity and several organic matters such as lead, manganese, cadmium. fluoride mercury, boron, etc. in the wastewater [2]. The management of wastewater systems is turning into a global issue to ensure a regular water supply and to avoid the over-exploitation of available water aquifers. The advancement of environmental technology has made wastewater not only manageable but also capable of being reused in beneficial ways such as for potable water supplies, groundwater replenishment, agriculture and irrigation, industrial purposes and environmental restoration [3]. Various water treatments that have been attempted and are conventionally being applied include chemical treatments, neutralization, adsorption, coagulation, mechanical treatments (e.g., sedimentation and Lamella settler), biological treatments, disinfection and membrane processes [4]. Among all of these, the membrane process is a very well-established separation process mainly in the water treatment industry due to the high selectivity [5], low energy consumption [6], moderate cost-to-performance ratio and high productivity [7].

A membrane is selectively permeable and is able to trap targeted substances aided by the driving forces such as the concentration gradient, pressure gradient, temperature gradient and electrical gradient [8]. Currently, membrane processes are evolving into membrane hybrid processes whereby they can be categorized into two kinds: (i) a combination of conventional processes with membrane process and (ii) a combination of two membrane processes [9]. In that case, the combination of adsorption/filtration to form adsorptive membranes for the removal of contaminants is being widely applied in water treatments, mainly for metal removal. In this regard, new adsorbents responsible for capturing contaminants and different fabrication technologies have been developed and applied to novel membrane synthesis and modification. Hence. novel membrane processes have been developed with improved membrane materials and designs. These major breakthroughs often lead to greatly enhanced process efficiency with permeate flux enhancement, fouling control and performance improvement.

Adding on to that, this study focuses on modifying the membrane structure to come up with a double-layer adsorptive membrane with a selective layer on top for boron removal from water using the co-casting technique. Adsorbing boron using membranes with adsorbents using the co-casting technique has not been developed so far but has been attempted for other water treatment applications [10]. The co-casting technique has shown promising results such as creating defect-free membranes in separating gases [11]. So, this serves as one of the motivations to improve the membrane structure for higher boron adsorptions. This write up will be devoted to introducing highly efficient adsorbents on the top layer for increased adsorption using the co-casting technique, and its advantages will be discussed further in this review with a boron nature. The limitations of existing boron removal technologies will also be studied. This review will be concluded with the challenges and future prospects in creating a highly adsorptive membrane for boron removal.

## 2. Boron Study

The average boron contents in the respective water resources are different due to the geochemical nature of the drainage area, the proximity to marine coastal regions and the imputes from industrial and municipal effluents [12]. In the readily available seawater treating applications, boron removal is conducted as the post-treatment procedure after desalination via Sea Water Reverse Osmosis (SWRO) membranes because they are efficient at removing charged species such as the borate ion rather than neutral molecules such as boric acid [13]. Desalination is a way of recycling seawater in order to obtain a steady supply of high-quality freshwater through the treatment of seawater and brackish groundwater. Figure 1 shows the evolution of desalinating technology as well as how changes have been made in terms of membrane technology and the desalinating process related to the removal of boron components.

The safe uptake of boron content in a human body is 20 mg, and food contributes greatly to the increment of boron content [23]. The risk to human health is through ingestion, which is through drinking, cooking and teeth-brushing. A boron level in water of more than 0.5 mg/L is perceived to be safe for bathing, hand-washing and dishwashing [24]. As shown in Figure 2, the boron limit in drinking water has undergone progress to result in a boron limit of 2.4 mg/L by the World Health Organization (WHO) [25]. As shown in Table 1, the tolerance limit of crops in absorbing the boron through soils or any water resources should be considered in order to use the treated water for irrigation purposes. This can be attributed to the reduction in the growth and quality of crops. Although the new guideline value is based on a human health perspective, some utilities may set the product water limits of seawater desalination plants to be as low as 1 mg/L to reflect agricultural-related issues [26].

### 2.1. Chemistry of Boron

Boron is a non-metallic element which falls under Group 13 of the periodic table. It is highly volatile and soluble [28]. Due to its lightweight property and higher ionization energy, boron forms covalent bonding rather than metallic bonding. Boron does not exist as a pure element. It is found in a bonded form [29]. Figure 3 shows the structure of boric acid, which shows hydrogen bonding. Each boric acid can form six hydrogen bonds, which are shown as dotted lines in Figure 3. When boric acid is dissolved in water, it forms borate ions. Equation (1) shows the dissociation equation of boric acid when it reacts with water.
(1)$$\;\;\;\;\;\;\;\;\;H_{2}O+B{\left(OH\right)}_{3}\to \;B{\left(OH\right)}_{4}^{-}+H^{+}\;$$

Boric acid acts as a weak Lewis acid to accept the hydroxyl ion of water and releases a proton into the solution, which forms an acidic solution. This is the reaction that makes the borate ion present in water resources. These borate ions can form complex and positively charged adsorbents through electrostatic reactions. In the water environment, boron generally exists as unionized boric acid with some borate ions.

The boron content in drinking water, seawater and wastewater is regulated in various countries because it is proven that higher amounts of boron consumption or exposure can be extremely toxic to living things. The adverse effects of boron have been reported when it is present in higher concentrations [29,30,31]. Generally, the attainability of boron in a solution is influenced by certain parameters such as the concentration, pH and temperature of the existing boron substances in the solution. Boron can be found as boric acid (B(OH)_3_) or in various forms of the borate ion (B(OH)_4_^−^) as a function of pH, as shown in Figure 4. However, the overbearing inorganic species in a solution is boric acid, which exists as small molecules at a neutral condition.

### 2.2. Sources of Boron

Boron enters the environment through natural and human-made processes. The natural weathering of soils and rocks releases boron into the water [33]. Currently, the demand for boron has increased due to the active development of modern architecture, which results in the evolution of the electronics, telecommunications (LCD screens), automotive, aviation and energy industries [34]. Boron is widely used in manufacturing fiberglass, detergents and bleaches [30]. Its highly volatile and soluble nature causes the boron waste from industries to be released into the atmosphere, forming boric acid to be combined in the water resources [35]. Figure 5 shows the origins of boron and the occurrence of pollution due to the presence of boron. Although nature has its own way of releasing boron, it is undeniable that human activities contribute to 90% of boron production [32,36].

### 2.3. Toxicity of Boron

High levels of boron exposure can cause serious health problems and can become extremely toxic to humans. Boron can be completely absorbed into the gastrointestinal and respiratory systems [31]. Boron is present in some fruits and vegetables, and it is a micronutrient needed by plants [37]. However, a higher dosage of boron consumption has been proven to affect living things. The major chronic toxicities are said to be developmental and reproductive when they are tested on several types of mice, rats and rabbits. Tetralogy studies conducted on them showed a reduction in fetal body weight and rib anomalies due to the uterus’ exposure to boric acid [38]. There were additional studies that reported about testicular toxicity in male rodents and decreased ovulation in female rodents [39]. These similar effects are assumed to take place within humans as well, since diet and drinking water are the most prevailing boron sources for them. Boron impacts skeletal metabolism by affecting the bone growth and compositional properties of soft tissues [40]. Its excessive amount is even related to arthritis and other inflammatory diseases related to the cardiovascular system [41]. The lifestyle in the Middle East and North Africa regions is one of the examples that shows the toxicity of boron, whereby irrigation water has been used for plant cultivation [42] and drinking [43]. Normally, when there is no downpour, there will be no chance for the boron to be leached out by rain, and it will continue to stay in soil and plants [33]. The over-absorption of boron affects the developmental and growth of plants, and this has a direct impact on its photosynthesis [37,44].

Apart from causing an increase in water scarcity, an excessive boron concentration in irrigation water, aquifers and soil has been reported to have adverse effects on crops such as limited crop yield and a reduction in the quality of productions [45]. Since boron is a nutrient for plants, the enrichment of nutrients in aquatic plants has become the main threat for water protection [46]. It can be assumed that the over-exposure of the boron nutrients can cause a rapid eutrophication process whereby they deplete the dissolved oxygen needed for aquatic animals. The accumulation of boron in water resources can be poisonous to other aquatic living organisms. Various studies have been conducted to analyze the impacts caused by boron exposure to fish and other water-living organisms [47,48].

### 2.4. Limitations of Available Conventional Boron-Removing Applications

The varying existence of boron at difference pH values in water leads to various attempts for it to be conducted commercially along with the academic laboratory scale. Generally, the act of removing boron from an aqueous solution revolves around two categories, namely, the sorption of pollutants onto solids and separation via membrane filtration. Some of the studies have proposed the integration of both categories. Under the sorption analysis, several mechanisms that are utilized for boron removal are chemical precipitation [49], ion exchange [50], adsorption [51,52,53] and electro-coagulation [54]. Table 2 shows the different boron-removing technologies with respect to the simplified explanation on the advantages and disadvantages of each conventional method.

Large sludge production with the poor handling of flocs in chemical precipitation and electro-coagulation makes them unattractive to be used for boron removal in wastewater treatment. In the adsorption process, the amount of removed boron increases with the dosage of adsorbent used; however, this issue can cause exhaustion during adsorption after a certain period. Therefore, the poor regeneration of the adsorbent tends to occur. Under the sorption analysis of boron, the stumbling block is chosen to be used for the ion-exchange method. Ion-exchange resins are only capable of removing boron in the ionic form.

As for the biological treatment, clean water that is produced through this method is limited. The single-step biological treatment is not suitable for removing too many of the pollutants from wastewater, whereas the conventional biological treatment has failed to reduce boron up to the standard limit in irrigation water [55]. For the Water Stabilization Ponds method, which uses microorganisms to remove boron from oil-field water, the biodiversity of ponds directly affects the degrading process because boron reduction is also connected with the Chemical Oxygen Demand in ponds [56,57]. Despite providing a low boron rejection, the electrodialysis method requires a high level of pre-treatment and frequent electrode replacement. Further, this method is also very costly.

Reverse osmosis can remove a fair amount of boron, even at high pH values in seawater desalination [58]. This is because the RO membranes are only capable of removing charged boron, which is dominant at higher pH values. A stronger repulsion between the membrane surface and borate ions causes the boron rejection to be enhanced. Several RO membranes that are operated at higher pH values have confirmed the occurrence of the scaling phenomenon (precipitation of salts on the membrane surface which can reduce the effectiveness and lifespan of the membrane) [59]. The removal of uncharged boric acid at low pH values still remains in its infancy because these molecules are small and can often pass through the RO membranes easily. In terms of commercially applicable membranes, it has been validated that the FilmTec membranes supplied by The Dow Chemical Company have several drawbacks too. The tighter RO membranes, which are designed for seawater treatment (seawater reverse osmosis, SWRO), show a high boron rejection ratio but a low recovery ratio and possess the need of elevated operational pressure. The looser RO membranes, which are designed for brackish water treatment (brackish water reverse osmosis, BWRO), show a lower boron rejection ratio but a slightly higher recovery ratio under lower operational pressure.

Several integrated methods such as polymer-assisted ultrafiltration (PAUF) and adsorptive membrane filtration (AMF) are used for boron removal. PAUF is capable of removing low concentrations of boron in streams, while AMF, which is equipped with boron-selective resins (BSR), is used in a traditional packed column arrangement. However, the commercially available BSRs are too large to be used together, and these BSRs appear to be costly [60].

The most effective way of eliminating dissolved boron species from seawater is through thermal desalination, which is implemented through multi-effect desalination (MED) [61]. It is a process of obtaining clean water from seawater after undergoing several stages of desalination. Although it is claimed to be an effective method of removing boron, this method has lost its commendation among several developed countries over the past few years due to the production of higher energy through the stages of distillation for seawater [62,63]. In addition, membrane distillation (MD) is one of the promising methods that offer a high retention of boron, which cannot even be achieved by most boron-removing methods [64,65,66]. However, the wetting and fouling issues that tend to happen due to the liquid entry pressure and feed temperature inhibit a better boron rejection, further making it unattractive to be practiced for boron removal [67].

Hence, the purpose of this review paper is to highlight the capability and potential of adsorptive membranes to replace the current conventional methods in removing boron efficiently. This review will also critically analyze the drawbacks of commercially available adsorptive membranes and single layered adsorptive membranes in terms of membrane stability and boron-removing performance. This paper will be a platform in introducing the technique of layering adsorbents in order to come up with a highly adsorptive dual-layered membrane for boron rejection. At the end of this review, the challenges and future prospects of enhancing the performance of adsorptive membranes in removing boron will also be emphasized.

**Table 2 membranes-12-00798-t002:** Advantages and disadvantages of different boron-removing technologies [68].

Conventional Boron Technology	Process	Advantages	Disadvantages
**1**. **Adsorption**	Process through which boron is adsorbed onto different kinds of solids.	Boron-selective No chemical consumption Availability of less costly natural adsorbents (e.g., coal, fly ash, red mud, etc.).	Cost of synthetic adsorbents Lack of regeneration studies Highly relies on the adsorbent’s performance Naturally available adsorbents are less efficient
**2**. **Biological treatment**	Application of biomass species in a single or combined process to treat water.	Feasible for removing boron.	Yet to be established and commercialized.
**3**. **Chemical precipitation**	Process of using chemical such as Calcium Hydroxide to form flocks and remove boron from water by sedimentation.	Simple Inexpensive Can provide higher boron removal	Sludge production Secondary pollution Consumption of chemicals
**4**. **Electrocoagulation**	Process of destabilizing suspended, emulsified or dissolved contaminants in an aqueous medium by introducing an electric current into the medium	Sludge settling Dewatering	Costly High chemical consumption
**5**. **Electrodialysis**	An electrochemical process causes ions to migrate through ion-selective semi-permeable membranes due to the attraction to two electrically charged electrodes	High separation selectivity	High operational cost due to membrane fouling High energy requirement
**6**. **Ion exchange**	Usage of boron-selective resins to remove and recover boron from water	High regeneration of ion-exchange resins Highly boron-selective	Costly
**7**. **Reverse osmosis**	A process where a partially permeable membrane is used to separate boron.	Less solid waste Less chemical consumption High efficiency Small space requirement	High capital and operational cost Operates at low flow rates Removal decreases due to scaling issues at higher pH values.

## 3. Adsorptive Membrane Technology

An adsorptive membrane is considered as a hybrid technology due to its dual functionality of adsorbing and filtering out pollutants. This membrane relies on a mass transfer process whereby the pollutants will get attached on the solid surface via chemical or physical interactions. Apart from offering flexibility in the design, this membrane is perceived to be cost-efficient due to its ability to regenerate adsorbents. The term “adsorption” was first coined in 1881 by a German physicist called Henrich Kayser [69]. It was further expanded in the mid-1980s [70]. Adsorption happens when separated or targeted ions are adsorbed on the membranes, when the solvent permeates through the pores of membranes that provide high flow rates and low internal diffusion along with faster adsorption and desorption rates [71]. Adsorptive membranes are highly utilized for the purpose of heavy metal removal [72], biotechnology-related separations [73] and other water and wastewater treatment applications [74]. This method is said to be simple and effective, as the adsorbent incorporated into the membrane matrix has the capability of capturing pollutants via chemical or physical relation.

Hybrid adsorptive membranes should be able to have an exclusive morphological structure and chemical structure in order for them to withstand the combined wastewater treatment applications. Therefore, their surface area and active sites in charge of adsorption must be very well preserved by adapting suitable fabrication techniques. As shown in Table 3, various fabrication techniques have been utilized in previous studies to attach multifunctional groups in a membrane framework in order to produce a highly adsorptive hybrid membrane for various contaminants. Each of the techniques has its own uniqueness in providing improved performance.

Adsorptive membranes are suitable to be used for boron removal as they retain boron molecules faster and provide a lower energy consumption and higher permeate flux. These membranes are easily functionalized to attach multiple functional groups as adsorption sites for favorable binding in order to maximize the adsorptive efficiency, as shown in Figure 6. The attachment of boron with a specific adsorbent can be described by a possible binding mechanism proposed in Figure 7. As shown in Figure 7, the -OH groups show a high affinity to boron, while these groups are not reactive to other elements. Adsorbents with these alcohol groups tend to bond through the formation of boric acid esters of boron or borate anion complexes with a proton as a counterion. Therefore, boron forms a coordination complex through a covalent attachment. As for the boron that exists as borate anions in water, it will be bonded with a positively charged adsorbent through ion exchange, and the structures are shown in Figure 8. Due to the release of acidic protons during complexation, the pH of the solution decreases. This complexation reaction also plays a vital role in boron uptake from water. The stability of the borate complex depends on the type of diol in the adsorbent. If the diol involves -OH groups being oriented in such a way that they accurately match the structural parameters required by tetrahedrally coordinated boron, a strong complex will be formed [82]. A stronger and stable complex will be able to provide efficient boron uptake.

### 3.1. The Mechanism of the Boron Adsorptive Membrane

Boron rejection via an adsorptive membrane is based on two mechanisms: rejection and adsorption. The boron contaminants are eliminated through molecular sieving and are then adsorbed by the attached adsorbents in a membrane matrix. In an ordinary membrane filtration, the molecules that are larger than the membrane pores will get retained when water is in contact with the membrane surface, thus allowing smaller solutes to pass through in order to provide pollutant-free water as the permeate. With adsorptive membranes, adsorbents are inserted into a membrane matrix as an active layer or even in a scattered version and are designed to couple with a preferable filtration mechanism such as Ultrafiltration or Microfiltration. Various functional groups that are in charge of boron rejection such as -COOH, -NH_2_ and -OH are inserted in the form of adsorbents to enable the rejection of boron solutes through ion exchange or surface complexation [83]. Apart from preventing the boron from passing through, the adsorbents will create a tight internal spherical complex and produce filtered water through an adsorptive membrane which can satisfy the required boron standards. The action of utilizing adsorbents through membrane filtration provides various benefits compared to the usage of adsorbents in batch processes. Adsorptive membranes possess the ability of easing the adsorbent regeneration process, as the boron will be eliminated in the form of sludge in the batch process. This is why adsorptive membranes are claimed to be environmentally friendly. The aligning of adsorbents on a membrane can also be a perk for the strength of the membrane, which has a direct impact on the performance as an adsorptive membrane. The advantages of an adsorptive membrane in removing boron can be described in terms of selectivity, permeability, hydrophilicity and mechanical strength.

#### 3.1.1. Selectivity

The adsorptive selectivity is crucial in quantifying the efficacy of an adsorption process regardless of the forms in which an adsorbent is utilized. It is based on the number of preferential locations on the pore space of an adsorbent for binding guest species such as boron. Boron adsorption happens in two steps: (i) affinity between the chelating group on the surface of an adsorbent with H_3_BO_3_ molecules, and (ii) physical attraction initiated by various forces such as Van der Waals, electrostatic attraction (between a protonated adsorbent and BOH_4_^−^ ions) or hydrogen bonding (bonding of -OH groups from H_3_BO_3_ molecules or BOH_4_^−^ ions with -OH groups), as shown in Figure 9. From Figure 9, the selectivity of boron through an adsorptive membrane also relies on the surface charge of the membrane. High positivity or negativity due to the surface charge of resultant adsorptive membranes facilitates electrostatic attraction and leads to higher adsorption of boron on the adsorption sites [72]. Logically, a more positively charged membrane surface can increase the number of negatively charged borate ions on the surface for further adsorption. However, the insertion of the UiO-66 nanoparticle via the implementation of a thin-film composite membrane shifted the zeta potential to be negative due to the hydrolysis of the acyl groups in providing -COOH. However, the boron rejection rate was very much enhanced [84]. This could be due to the ability of the negatively charged membrane surface to draw more water molecules to the membrane surface, which was followed by the presence of many boron compounds.

#### 3.1.2. Hydrophilicity and Permeability

The permeability is correlated with the hydrophilic nature of the membrane, which affects the movement of water molecules in the membrane matrix. Hydrophilicity is the ability of a specific material to draw water molecules, which can expand water channels in the membrane matrix. Most adsorbents that are used for boron removal, such as resins [85], metal oxides [85], nanotubes [86] and the metal organic framework [87], are hydrophilic. Hydrophilic materials in a membrane matrix provide the ability for water molecules to be spread across, thus maximizing the contact. This can be related to membrane fouling, as the plugging of pore openings in the structure of a membrane can cause surface contamination, which leads to the deterioration of the membrane performance. With hydrophilic adsorbents filling up the pores of the membrane, the water permeation is enhanced due to the function of adsorbents as the water carrier throughout the membrane process. The increase in permeability also happens due to an increase in the polymer network free volume and the formation of cavities in the nanoparticle–polymer interface [88,89]. This is why adsorptive membranes are perceived to be suitable for boron removal, as they can provide better rejection without hampering the water flow. To make this achievable, it is important to come up with the optimum adsorbent amount that can be tolerated by a boron-adsorptive membrane. The rapid adhesion and deposition of adsorbents on the membrane due to the exceeding optimum value can cause pore blockage and reduction in the boron rejection.

#### 3.1.3. Mechanical Strength

An adsorptive membrane boasts better mechanical features in order to tolerate the pressure during water treatment. Sufficient mechanical strength is necessary to define the stability, end use and processability of the fabricated membranes. The interaction between the adsorbents and polymer components in an adsorptive membrane contributes a significant role in imparting mechanical properties [90]. The well-dispersed adsorbent in a polymer matrix restricts the chain movements and thereby supports mechanical strength. However, the variation in mechanical properties depends on the size, uniformity and volume fraction of the incorporated particles [91]. This is because the aforementioned criteria play an important role in the porosity of an adsorptive membrane. Increased porosity can cause a higher tendency for pore aggregation, which makes the membrane fail [92]. This is due to the increment in the strain area around the pores and is followed by uneven stress distribution [93]. Membranes with circular pores are able to possess higher mechanical strength compared to uneven pores. Incorporating an adsorbent into the polymer matrix was able to fill up voids and provide a denser structure, which helped in strengthening the membrane structure. Wang et al. reported that nanofiber membranes without surface modification have shown a higher elongation break compared to membranes that are inserted with the vicinal hydroxyl groups [94]. This may be due to the interaction that exists between polymer groups and hydroxyl groups, as they provide a tight holding that prevents them from any deformation.

### 3.2. Effective Operating Parameters Responsible for Boron Adsorption by an Adsorptive Membrane

Once the adsorbent is in the vicinity of the membrane surface, it is held together by the combination of several forces such as the electrostatic force and mechanical attachment with the polymeric surface structure. The attachment of the adsorbent with the membrane structure is mostly reversible in certain studies. The adhesion of the membrane structure with an adsorbent plays a vital role to enable the secured adsorptive processes of contaminants by the adsorbent in a membrane. Apart from structural suitability, there are other key parameters that are in charge of the adsorptive membrane process, such as pH, temperature, contact time, adsorbent concentration and the initial solute concentration that is to be adsorbed.

#### 3.2.1. Effect of pH

The effect of pH plays a major role in determining the adsorption performance of the adsorbent. The mechanism of boron adsorption is mainly reflected by the surface charge of the adsorbent at various pH values of the adsorbate solution. As reported by the boron adsorption removal study by commercial and modified activated carbons, the maximum boron adsorption was found at a pH value of 8.5, and the boron removal did slightly decrease when the pH value was lowered from 7 to 3 [95]. Another study using RO membranes showed more than 99% of boron removal at a higher pH value [96]. This is due to the electrostatic repulsion between the negatively charged RO membrane and the borate ions. The boric acid solution dissociates into a negatively charged borate ion at higher pH values, causing more boron to get rejected.

#### 3.2.2. Effect of Temperature

By varying the temperatures used for the adsorption process, the nature of the process can be determined either exothermically or endothermically. Generally, the adsorption capacity of the adsorbent increases as the temperature of the feed solution increases. This is because, with increasing temperature in the circulating water, the rapid movements of borate ions in a solution are triggered, resulting in a higher kinetic energy. Therefore, the movement rate of borate ions to the surface of the adsorbent accelerated to increase the adsorption capacity of the adsorbent for boron [97]. This can be supported by the result from the removal of boron from wastewater by ion exchange in a continuous reactor carried out by A.E. Yılmaz et al. [98]. The removal of boron from wastewater increased with an increasing temperature up to 313 K.

#### 3.2.3. Effect of Contact Time

Studies on the contact time of the adsorbent and adsorbate are important to predict the reaction rates and adsorption equilibrium of the reaction. The effect of contact time on boron adsorption was investigated using gluconate-intercalated hydrotalcite [99]. The adsorption performance increased with the contact time and reached equilibrium after 4 h.

#### 3.2.4. Effect of Adsorbent Concentration

Most boron adsorption studies reported adsorbent concentration as one of the factors affecting boron capture. An adsorption study of boron using fly ash was conducted in batch and column studies [100]. The study revealed that the adsorbed boron percentage increased with increasing adsorbent dosages. The removal efficiency increases up to an optimum dosage, beyond which the removal efficiency is negligible. Another study involved nanofiltration thin-film hollow fiber membranes with adsorptive bentonite and LDH nanoclays, which were tested for boron removal [101]. The boron removal increased steadily with the number of adsorbents inserted in the membrane for both bentonite and LDH nanoclays. It can be concluded that the improvement of adsorption capacity was due to the increment of the available active site for boron adsorption at a higher loading of the adsorbent into the membranes.

#### 3.2.5. Effect of Initial Feed Solute Concentration

The amount of the substance being adsorbed varies with varying initial solute concentrations and increases when the solute concentration is higher; eventually, it becomes constant due to the driving force offered by the increased solute concentration, which is sufficient enough to overcome the resistance to mass transfer between the solid and liquid phases [102]. So, the adsorption will enhance with higher initial concentrations. By studying the adsorption capacity of the adsorbent by varying the initial solute concentration, the relationship between the amount of adsorbed solute at equilibrium and the equilibrium solute concentration could be revealed. Previous studies related to boron showed that the adsorption capacity increased with the initial concentration until an optimum value was reached. A study on determining the capability of unmodified rice husk in adsorbing boron showed that the percentage of boron increased with increasing concentrations of rice husk [103]. According to D. Kavak in his study [104], the effect of the initial concentration of the adsorbate on the percentage of boron removal can be explained in terms of the number of exchangeable sites in the adsorbent structure and boron-to-adsorbent ratios. The decrease in boron adsorption happens as the adsorbent-to-adsorbent ratio increases; the adsorbents become saturated enough and limit the adsorption process.

### 3.3. Types of Adsorptions Associated with Boron Removal

The adsorption of boron via membrane adsorption depends on the nature of the surface attachment due to adsorbents. Physical adsorption (Physisorption) is caused by Van der Waals forces and is the easiest to be separated because these forces are weak in nature [105]. It does not depend on the nature of the adsorbent or adsorbate. For instance, the retention of boron molecules on the pores of a membrane is because of the different sizes as they are held onto an adsorbent or the surface of a membrane by weak intermolecular forces. This process can be easily reversed. Secondly, chemical adsorption (chemisorption) takes place through chemical bonding between the boron and adsorbent [106]. Boron is chemisorbed on the adsorbent surface and is difficult to be removed due to the electron sharing or ion exchange. The process can possibly to be reversed, but the regeneration of the adsorbents will require higher energy. Under favorable conditions, both procedures can happen simultaneously or alternately. Thirdly, exchange adsorption is caused by the charge attraction between the adsorbate and the surface [107]. This will occur when a positively charged adsorbent is used to attract the negatively charged borate ions from aqueous solutions.

### 3.4. Commercial/Published Work for Boron Removal by Adsorptive Membranes

Boron removal from various water resources has been successfully attempted by utilizing proper functionalized adsorbents along with suitable filtration mechanisms. From Table 4, the boron-removing adsorbents are used as adsorptive membranes in order to satisfy the requirements of the boron standard based on the purpose and source of the water.

RO is the most common method of purifying seawater in order to obtain drinking water in various parts of the world. The limiting capability of RO membranes to remove uncharged boron has been widely discussed in various boron-related studies. At a pH > 9, where the boron exists as borate ions, RO membranes have shown a good boron-removing ability, which is caused by the lack of electrostatic repulsion on the membrane surface. Therefore, commercial RO membranes have been tested in terms of boron rejection at a laboratory scale. Ruiz-García et al. figured out the boron rejection using two commercial membranes (TM820S-400 and TM820L-440) provided by the Toray manufacturer at different operating parameters [108]. Experiments towards commercially available membranes were upgraded by forming an integrated system. For example, X. Bao et al. investigated the transportation of salt and boron removal by commercial SWRO membranes via an electrically assisted RO process [109]. The RO related membranes used for boron removal shift towards being adsorptive membranes with the functionalization of boron-capturing adsorbents in the membrane matrix. As an example, a thin-film composite membrane layer is functionalized using the surface grafting method, which is used for RO purposes for seawater [110]. In this work, 3-amino-1,2-propanediol (APD) and tobramycin (TOB) were grafted on the surface of the commercial RO membrane SW30XLE to increase boron removal and fouling resistance. Besides providing a higher salt and boron rejection, this type of membrane provided excellent antifouling properties, including organic fouling resistance, scaling control and bactericidal capacity. This was the stepping stone to come up with a fabrication of high-performance RO membranes with enhanced perm-selectivity and antifouling properties, which are greatly desired in SWRO desalination. Metal Organic Frameworks (MOF) such as UiO-66 nanoparticles have shown excellent boron removal abilities in batch studies on removing boron [111]. Their regenerating ability is quite good. Liu et al. incorporated these nanoparticles into a thin-film nanocomposite RO membrane for boron removal [84]. The study successfully formed boron-captivating functional groups such as Zr-O, C-H and -COOH. The increase in water flux can be explained by the creation of preferential pathways for water passage due to enhanced porosity. Increasing the filler loading beyond the optimum concentration led to reverse results of decreased flux. The performance of the TFC membrane was 11% higher than the trademark fixed for a pristine TFC membrane.

Adsorbing resins such as chitosan is observed to affect the charge of the membrane greatly, and they captivate pollutants by several mechanisms such as chelation, electrostatic attraction or even interchanging ions between the adsorbent and pollutant. However, the mechanism highly depends on the pH values, solution composition or pollutant behavior. Chitosan has been introduced in membranes for removing various metals such as silver [112], copper ion [113] and lead [114] with other hazardous metal ions [115]. Due to its biological and chemical properties such as non-toxicity, biocompatibility and high chemical reactivity, chitosan has also been used in adsorptive membranes for boron removal [116]. The presence of amine and hydroxyl groups in chitosan can enhance the affinity towards metal ions [117]. There were also studies that highlighted the shortcomings of pure chitosan to be used for adsorption because they tend to have poor mechanical strength and chemical stability [118]. These shortcomings can be tackled by blending chitosan with other polymeric materials. A high-performing low-energy RO membrane was fabricated, whereby the polyamide layer was modified with the chitosan polysaccharide to be used for boron removal [119]. N-methyl-D-glucamine-functionalized adsorptive membrane filtration systems were used [120]. According to Du et al., the polyol-grafted polysulfone membranes were able to provide the amine group, the high vicinal diol content and the branched structure, which are responsible for the complexing activity in boron uptake [121].

Molecular plugs such as aliphatic amines, polyisobutylene and 4-nitrobenzenesulfonyl chloride are able to increase boron rejection [122]. An interesting attempt is to incorporate sulfonyl molecular plugs in the polyamide network in order to induce a swelling capability. The 4-nitrobenzenesulfonyl chloride (NBS) molecules that were embedded into the pores of RO membranes were able to capture even small and neutral boron molecules from seawater [123]. This was due to the introduction of -SO_3_H groups, which hindered the diffusion of boric acid in the membranes. Hu et al. emphasized in his paper that the availability of amine monomers is limited in seawater desalination applications and that most high-performance SWRO membranes are still prepared from m-phenylenediamine (MPD)-based monomers [124]. To overcome this limitation, the researcher came up with a charge aggregate-induced reverse osmosis membrane to remove boron, whereby a new sulfonated diamine monomer, 4,4′-(1,2-ethane-diyldiimino)bis(benzenesulfonic acid) (EDBSA), was synthesized and used as the sole amine monomer for the membrane. The highly aggregated charges display strong interactions and an excellent complexation effect with boric acid. This study was a solution to removing boric acid at a neutral condition, which was not achieved by most of the studies on boron removal. This is because the molecules are too small and can easily pass through the pores of the membranes. This invention was compared with a lab scale MPD-based membrane, showing a better durability and a capability of maintaining its reasonably good performance.

Inorganic materials that are incorporated with adsorptive membranes not only have excellent thermal and chemical stability but also show great potential when used in water desalination processes [125]. Nanosized metal oxides are widely used for the removal of heavy metals from water since they can provide a larger surface area and an increased affinity for the process to take place [126]. The combination of nanoparticles along with other inorganic substances was carried out as well. A comparison was made between iron oxide-hydroxide-based nanoparticles and nanosized iron oxides-impregnated activated carbon for boron removal [127]. Apart from supporting the combination of two inorganic substances, this study also successfully came up with a greater recovery of boron. Attaching inorganic elements to be used as a membrane was applied in the removal of other heavy metals [125]. Polyethersulfone flat sheet membranes that were incorporated with titanium dioxide (TiO_2_) nanoparticles to be functionalized with multi-walled carbon nanotubes were fabricated using the wet phase inversion method. The mixed matrix membranes with 0.7 wt% revealed a higher boron rejection [128]. The spreading of adsorbents in a membrane matrix was widely applied for the adsorption process of boron. The accumulation of too many inorganic particles could form larger nanoparticles in the membrane matrix, making the adsorption process complicated, which can lead to membrane failure [123]. This was because this phenomenon could reduce the exposure of incorporated adsorbents to boron.

Furthermore, one of the issues in removing boron by electrodialysis is its ability to be effective in alkaline conditions in removing borate ions rather than removing the dominant species, which is the non-charged B(OH)_3_. To overcome this shortcoming, Sun et al. adopted a bipolar membrane electrodialysis (BMED) and incorporated graphene oxide-modified porous P84 co-polyimide membranes for boron removal [129]. The physicochemical properties of the anion exchange membrane were very much improved with the quaternized adsorbents. Despite having a higher boron separation, this invention was able to provide lower energy consumption along with excellent BMED performance.

**Table 4 membranes-12-00798-t004:** Previous studies about boron removal by adsorptive membranes.

Adsorptive Membrane	Membrane Configuration	Source	Initial Boron Concentration (mg/L)	Adsorption Capacity (mg/L/mmol/g)	Boron Removal (%)	Water Flux	Reference
**N-methyl-D-glucamine-functionalized adsorptive membrane**	Hollow fiber	Saline geothermal water	11.0 mg/L	0.52 mg/L	95.3	-	[120]
**3-amino-1,2-propanediol (APD) and tobramycin (TOB)-grafted commercial SW30XLE RO membrane**	Flat sheet	Seawater	-	-	92.2	33.4% improved flux	[110]
**Polysulfones-grafted polyol polymers**	Flat sheet	-	5 mg/L		-		[121]
**Monomer 2,3-dihydroxypropyl methacrylate (HPMA)**				0.20 mmol/g		70 L/m^2^ h	
**Monomer 3-(N-glucidol-N-methyl)amino-2-hydroxypropylmethacrylate (GMHP)**				0.44 mmol/g		90 L/m^2^ h	
**Monomer 2-(bis(2,3-dihydroxypropyl)amino)ethyl methacrylate (HAEM)**				0.46 mmol/g		80 L/m^2^ h	
**Hybrid PVDF-PVP membranes with nano TiO_2_ as an additive**	Hollow fiber	Leachate	8.2 mg/L	0.43 mg/L	94.75	223 L/m^2^ h	[123]
**Charge aggregate-induced RO membrane-4,4′-(1,2-ethanediyldiimino)bis(benzenesulfonic acid) (EDBSA) with trimesoyl chloride (TMC) on a poly(ether sulfone) (PES) substrate**	Flat sheet	Seawater	5 mg/L	0.5 mg/L	90.6	8.5 L/m^2^ h	[124]
**Thin-film nanocomposite RO membrane with UiO-66 nanoparticles**	Flat sheet	Brackish water	5 mg/L	-	99.08	56.83 L/m^2^ h	[84]
		Seawater	5 mg/L	-	99.27	61.32 L/m^2^ h	[84]
**Graphene oxide-modified porous P84 co-polyimide membranes**	Flat sheet	-	100 mg/L	-	76.6	-	[129]
**Polyol-functional polysulfone membranes**	Flat sheet	-	300 mg/L	1.61 mmol/g	-	45 L/m^2^ h	[130]
**Hyperbranched-polyol-tethered poly(amic acid)electrospun nanofiber membrane**	Hollow fiber	-	5 mg/L	5.68 mmol/g	-	-	[94]
**Thin-film composite with a phosphonic acid derivative of TiO_2_**	Flat sheet	Seawater	5 mg/L	-	96	38 L/m^2^ h	[131]
**Polysulfone membrane with an amphilic graft glycopolymer**	Flat sheet	-	300 mg/L	0.193 mmol/g	-	475 L/m^2^ h	[132]
**Varied Polytetrafluoroethylene (PTFE) micro powder-added optimized PVDF nanofiber-based membrane distillation**	Hollow fiber	Geothermal water	60.84 mg B/L	0.5 mg B/L	-	27.7 kg/m^2^ h	[133]

## 4. Single-Layered Adsorptive Membranes

Single-layered adsorptive membranes have always been the top choice in removing boron since its fabrication process is easier. It is the idea of filling a whole membrane matrix with adsorbents to be used for further rejection testing. This method has been developed in order to solve the hydrophobicity issue that has been faced by most polymeric membranes. The failure of possessing a low affinity towards water makes most of the membranes unfavorable for any water treatment applications. Regardless of the materials used, the overall aim of an adsorptive membrane is to have an excellent rejection without compromising the stable flux and mechanical strength of the membranes. Pore size is one of the important factors that contributes to the aforementioned standards. Incorporating adsorbents into a polymer material can contribute to the narrow pore distribution, which is needed in industrial application nowadays. However, the loading of the adsorbent in an adsorptive membrane must be carefully studied, as too many particles in a polymer matrix can cause aggregation, which can further deteriorate the membrane performance through a cakey layer formation. Due to these reasons, single-layered adsorptive membranes tend to have a shorter lifespan, as they need to be replaced frequently with other membranes for better membrane performance. Therefore, an upgraded version that focuses more on the mechanical strength, which is the double-layer adsorptive membrane, is developed without giving up any of the qualities exhibited by a single-layered membrane. Table 5 shows the overall comparison of both single-layered and double-layered adsorptive membranes in terms of structure, benefits and common issues.

### 4.1. Methods of Incorporating Adsorbents

According to the preparation method, there are various ways in which adsorbents can be inserted into a membrane matrix, as shown in Figure 10. Positioning plays an important role in the adsorptive process of an adsorptive membrane. Single-layered membranes are one of the most common adsorption membranes. Their preparation process is simple, but it is easy to agglomerate when the adsorbent content is high. Adsorbents that are embedded inside the polymer matrix have a lower adsorption capacity and a longer adsorption equilibrium time [118]. There are several fabrication methods that are already in practice for adsorptive membranes such as coating, grafting, assembling, composite membranes, imprinting, phase inversion/solution casting and electrospinning.

One of the techniques to come up with a highly adsorptive membrane is through modifying the surface of a membrane, as its exposure to pollutants plays a vital role in the process. Blending and coating provide physical adsorption whereby layers with increasing affinity towards adsorbates are formed without any chemical interaction. It is done by dispersing the adsorbents into the solvent via an ultrasonic bath or by stirring where the polymer is added. Then, the solution is cast on a flat surface and dried by an evaporating solvent. The coating process can be categorized as dip coating, spin coating and spray coating. This method can promote a layer-by-layer assembly for the adsorption process. Figure 11 shows the illustration of various coating techniques, which are (1) dip coating, (2) spin coating and (3) spray coating. Dip coating is a method of depositing functional material on a membrane substrate. It is done by immersing the membrane substrate into a coating solution for an effective formation of layers [124,134]. Spin coating is a technique of spreading a uniform layer onto a solid surface by a centrifugal force and requires a liquid vapor interface [135]. Additionally, the coating that is applied on an object with a liquid spray is called spray coating [136]. One of the studies involved modifying a polysulfone (PSF) membrane via co-depositing polyethyleneimine (PEI) with dopamine (DA) in one step to produce amine-rich surfaces. The functionalized membranes further react with glycidol for boron removal applications [130]. The motive of this study was to introduce hydrophilic properties on the membrane surface, which could aid in the eliminating process. The fabricated membranes had a very strong hydrophilicity, but the water flux decreased in the process. Therefore, this study suggested this method as a potential attempt to be applied in boron removal applications by improving the permeability factor. Other reported drawbacks of this method include the longer process time for dip coating due to diffusional resistance and rinsing for the elimination of polyelectrolyte complex formation, which can lead to flocculation. Although spin coating is claimed to be the fastest, the surface area of a material to be coated is not sufficient. For spray coating, the possibility of the coating material being drained due to gravity is higher [137].

Another method of surface modification is grafting, whereby a pollutant is adsorbed onto the surface that is functionalized with an adsorbent by chemical adsorption such as ion exchanging or electrostatic reaction. Du et al. synthesized a series of polysulfone membranes that were grafted with polyol polymers in order to test the removal of boric acid [121]. The attached ligand structure with a high vicinal diol content and the presence of the amino group was able to increase the complexing efficiency for boron uptake. Although the surface hydrophilicity was enhanced, the decrease in water flux was very obvious in this study. Apart from that, the poly(amic-acid) electrospun nanofiber membranes that were grafted with hyperbranched polyols were able to provide good regeneration as an adsorbent, besides showing an outstanding removal of boron [94]. Surface grafting has several advantages, which include the ease of the modification process, the higher chemical stability and the low delamination of grafts [138]. However, there were some studies that showed changes in the pore structure that could disturb the water movement in membrane-based water applications [139,140].

Boron removal has been widely tackled using composite membranes. There has been significant research on whether micro and nanoparticles are suitable to be used for boron adsorption and on their potential when being used as composite membranes [141,142,143]. A composite membrane is the combination of immiscible additives with a polymeric substance. They are used to produce membranes with a high adsorption capacity, fast kinetics, reduced fouling, promising reactivity and flexibility. Composite membranes are fabricated by adding micromaterials or nanomaterials in the structure of a membrane on the surface or dispersing in the matrix. In that context, thin-film composite membranes have received increased attention in water desalination applications since the scope can always be widened with available materials in nanotechnology. Thin-film membranes are able to perform well compared to asymmetric membranes in desalination applications [144]. Thin-film composites have an active layer which is rich in nanoparticles to provide easier adsorption, whereas the classic composite membranes tend to have agglomerated particles in the polymer matrix [145]. Kumar et al. functionalized titanium dioxide (TiO_2_) nanoparticles with phosphonic acid to remove boron from seawater [131]. Apart from having a higher boron rejection than commercial membranes, the study of removing neutral boric acid was a success due to the pore narrowing effect.

The limited affinity of removing metals towards adsorbents is one of the critical issues that is raised during the fabrication process of adsorptive membranes. The method of imprinting can be solved with the application of specific binding sites that are incorporated into the membrane matrix. Imprinting is about introducing synthetic receptor locations in the membrane matrix that recognize, remember and identify the target species among others in a solution. This technique is used to obtain selective membrane adsorbents in order to overcome the issues in selectivity that result from the limited specific binding capacity. It is carried out via the addition of specialized configured voids to the polymer by inserting the target when preparing the membrane and then immediately leaching it to vacate the active sites. Imprinted membranes are created in both flat sheet and hollow fiber configurations [146]. So far, this method has not been applied for boron removal; however, this has already been in practice for other metal removal applications [147,148].

Phase inversion/solution casting is the most common method of membrane fabrication nowadays. It is a process of membrane synthesis using a polymer-solvent mixture to form a homogeneous solution at specific conditions of temperature and composition, which can be differentiated with a slight change in the conditions. This method can give a better dispersion of fillers, an excellent interaction between the matrix and the filler and a uniform merging of the polymer and adsorbent. It can be done by evaporating a volatile solvent from the homogenous solution or cooling a casting solution. Phase inversion can entrap nanomaterials within the matrix, where they can get blended and dispersed in a polymer dope solution. Shi et al. prepared an amphiphilic graft glycopolymer and used it as membrane additives to prepare hydrophilic polysulfone membranes [132]. They were prepared using phase inversion with different weight ratios of additives and were then cast as flat sheet membranes. The hydrophilic glycopolymer segments were observed to be accumulated at the membrane surface in phase inversion when blending with polysulfones. This method was able to provide good antifouling of the membrane and boron complexing properties [132]. The main drawback of this technique is the huge amount of solvent wastage during preparation. Further, it is difficult to control the precision and the uniformity of the prepared membranes [149].

Furthermore, electrospinning is a high voltage-driven process that creates an electric field to induce electrostatic repulsion forces, which shatter the polymer surface tension and stretch its droplets to form solid continuous nanofibers. This method is used to synthesize nanofibrous membranes with an improved efficiency and excellent removal capacity for heavy metals and organic pollutants. This method was prepared by using a pump that was equipped with a nozzle-fitted syringe, a spinneret, an electric current source and a counter electrode or grounded target. The use of high voltage creates an electric field and droplets at the nozzle. When the charged jet accelerates towards the collector, the solvent evaporates and forms the nanofiber. Ozbey-Unal et al. fabricated a hydrophobic nanofiber membrane via electrospinning to be used for removing boron from geothermal water [133]. The motive of the aforementioned study is to prevent membrane wetting by narrowing the open fiber structure of the membrane and improving its hydrophobicity via coupling heat treatment. The driving force is also enhanced to aid boron and salt removal.

Single-layered adsorptive membranes have been the new-generation technology in water treatment applications since they combine the inherent characteristics of polymers and adsorbents. Most of the aforementioned methods have been focusing on these types of membranes. Previous research focused on the chemical and structural form of adsorbents. There are very limited studies regarding the placement of adsorbents in the membrane. Surface-functionalized adsorptive membranes tend to have a high adsorptive capacity and a short equilibrium time. However, surface-deposited adsorptive and surface-assembled adsorptive membranes have the detachment risk of deposition and assembly layers during application and regeneration. However, these detachment issues can be easily tackled by several techniques such as the co-casting method when fabricating by the dual-layered membrane.

### 4.2. Issues/Problems Found in Single-Layered Adsorptive Membranes

Studies regarding single-layered adsorptive membranes have raised some serious issues associated with membrane fouling. One of the prominent limiting factors is the dispersion of the adsorbent onto a polymer matrix. Studies suggested that the aggregation or dispersion behavior control, which is the first process for the preparation of new functional materials that incorporate nanoparticles, is very difficult for nanoparticles that are less than 100 nm in diameter due to surface interactions [150]. Several studies have shown agglomeration issues, whereby adding more adsorbents into a single-layer polymer matrix will cause more agglomeration, which can lead to membrane fouling [151]. The over-attachment of solutes onto the membrane surface or into the internal structure of the membrane could form an additional barrier or block the membrane pores by preventing the solvent from transporting through the membrane, which could raise the trans-membrane pressure and lower the permeate productivity [150]. Some fouling materials even destroy the membrane and shorten its service life. Several techniques should be tackled from time to time in order to prevent this phenomenon from happening [152].

## 5. Dual-Layered Adsorptive Membranes

The difficulty and higher cost of merging the advances of materials with the fabrication of the membrane have created the idea of dual-layered membranes. Dual-layered membranes are membranes with two layers which consist of different materials. The sub-layer is a porous structure which is responsible for providing mechanical support, while the top layer is made from a high performance which has a selective dense material.

### 5.1. Various Methods of Fabricating Dual-Layered Membranes

Asymmetric membranes are characterized by a thin and dense skin layer on top of a porous structure. The thin layer acts as a barrier for substances to move in and out of the membranes, despite providing a stronger mechanical strength. Dual-layered membranes fall under thin-film composite membranes. Recently, the development of this type of asymmetric membrane has been at its peak due to its interesting properties. It must be mentioned that each layer (the top selective layer and the bottom porous substrate) of this type of membrane can be independently controlled and optimized to achieve the desired selectivity and permeability while offering excellent mechanical strength and compression resistance. There are several fabrication methods that are applied in order to come up with high-performing dual-layered flat sheet-type membranes such as the interfacial polymerization [153], layer-by-layer [154], coating, cross linking [155] and co-casting techniques [156]. Table 6 shows the various methods utilized in developing dual-layered membranes with their respective application in treating water. Dual-layered flat sheet membranes have also been fabricated in previous studies to remove boron, as listed in Table 7. So far, there are very limited studies that have been attempted for boron removal, especially for double-layered flat sheet adsorptive membranes for boron removal. Boron has a special chemistry, and, thus, it mostly differs from that of other trace elements. In this regard, boron chemistry depends strongly on pH and ionic strength. Therefore, boron studies are mostly focused on having good adsorbents, which could satisfy these parameters because their high surface-to-volume ratio provides higher adsorption. The possibility of having a reduced adsorption of boron in adsorbents when they are being utilized as double-layered membranes could be something researchers fear [157]. This is because of the probability of the functional groups of the adsorbents reacting with the polymer, especially where polar functional groups are present. This might be a stumbling block for them progressing further as adsorptive membranes.

### 5.2. The Proposed Co-Casting Technique

Co-casting is proven as an effective one-step membrane fabrication technique where the active top layer is extruded simultaneously with a supporting layer, which results in a composite double-layered membrane structure [160]. Commonly, two different polymers are employed in the preparation of composite membranes so as to improve the membrane permeability and selectivity as well as to decrease the cost in membrane materials. It is well known that the miscibility of the two membrane materials plays an important role in their adhesion or delamination. For example, well-miscible polymer pairs often lead to good adhesion; therefore, composite membranes have been prepared with the same or similar polymers in both coating and support [161,162].

Co-casting is a method whereby a coating solution is casted simultaneously with a support solution, resulting in a DL membrane structure. The universal co-casting method utilizes two individual cylindrical-shape casting knives. Each side of the knife can cast membranes with different thicknesses (100, 150, 200, 250 μm) [163]. As a result, the DL membrane thickness can be adjusted, and the interface between the two layers can be improved. Therefore, the delamination/adhesion effects can be improved using the co-casting technique, which can result in stable membranes. Surprisingly, the co-casting method for fabricating membranes for boron removal has never been attempted so far. The illustration of this method is shown in Figure 12.

#### 5.2.1. Parameters Involved in the Process of Co-Casting

When it comes to the co-casting technique, there are several factors that affect the process, as listed below.

##### Solvent and Non-Solvent Selection

The first step was to form a homogeneous solution by choosing the solvent to dissolve or easily disperse the polymer. The quality of the solvent has a direct impact on the membrane morphology. A weak solvent produced a sponge-like porous structure [164], whereas a stronger solvent led to the formation of macrovoids [165]. It was also important to make sure that the solvent and non-solvent were miscible such that a more porous membrane could be formed when there was a higher affinity between these two solvents. A low relative miscibility between the solvent and non-solvent could delay the de-mixing process between these two solvents.

##### Polymer Concentration and Properties

Since the polymer is the component that forms the membrane matrix, the polymer concentration in the casting solution will influence the final membrane morphology. Generally, a higher polymer concentration generates a lower gravimetric porosity. When the polymer concentration is above a certain threshold, there is not enough solvent and non-solvent exchange in the dope solution to form the pores during the phase separation and solidification process. Therefore, the membrane gravimetric porosity is lowered, and the permeability is lost. The blending process supports the formation of macrovoids, which results in a much better performance.

##### Additives in the Polymer Concentration

Additives clearly improve the pore formation and structure, enhance the pore interconnectivity and increase the hydrophilicity, which has direct impact on the delamination phenomenon of the dual-layered membranes [166]. Several works of research revealed that adding hydrophilic polymers such as PVP led to a thicker membrane skin layer, whereas the sub-layers of membranes had dense structures with fewer macrovoids, and the presence of finger-like-structures also disappeared [167,168]. The addition of surfactants such as sulfonic groups or carboxyl groups can produce large pores on the top surface, and a more porous structure in the sub-layer leads to thicker membranes. The delamination issue can be resolved by adding additives [161].

##### Film Casting Conditions

The main film casting conditions are the composition and temperature of the coagulation bath. The addition of small amounts of solvent in the coagulation bath is an effective method to prepare a non-porous membrane because it can reduce the rate of mass transfer between the non-solvent and the casting solution, which can provide instant de-mixing [169]. The addition of a very high concentration of solvent will inhibit the dilution effect from forming a good polymeric membrane [166]. Apart from that, the casting temperature should be taken into consideration too. This is because the viscosity, µ, will be affected, and this can cause changes in the membrane structure and internal morphology [170]. There are some conditions analyzed by Hashemifard et al. to enhance a better adhesion between the top and bottom layers: (1) casting the sub-layer (SL) dope with a concentration that is high enough to satisfy the condition, µSL > µTL (µ = viscosity); (2) casting the top layer (TL) at a temperature higher than the sub-layer temperature; (3) incorporating a certain amount of filler in the sub-layer dope. All these conditions were able to avoid the detachment between the two layers [160].

Li et al. investigated the factors that control the adhesion or delamination in the following aspects [162]: the coagulation value, the interpenetration of polymer chains, the shrinkage difference in layers and the adhesive components. It was found that the shrinkage values are important in determining adhesion and delamination. At higher coagulation bath temperatures, the membrane shrinkage of PEI and Polysulfone (PSf) membranes in the thickness was much closer, and a good adhesion was observed at temperatures above 65 °C.

So far, co-casting methods have not been introduced in applications of boron removal, but this technique has been already attempted in gas adsorption [11]. The defect-free selective layer formed in this method providing high permeability and selectivity is the perk of this method for higher boron adsorption. This review serves as a stepping stone in generating more creative ideas associated with co-casting adsorptive membranes for boron removal. This is because these parameters can be easily controlled, and the possibility to generate a denser layer of adsorbents for boron adsorption is higher.

### 5.3. The Advantages of Bilayer Membranes by the Co-Casting Technique in Ensuring the Boron Adsorption

Bilayer membranes come up with the porous substrate film layered beneath a dense boron-rejecting layer. The major advantage of having the two layers made of different chemicals is that each layer can be individually synthesized or customized so as to optimize the overall performance of the membrane. Compared to the common mixed matrix membranes for the adsorption process, these bilayer membranes are said to have better salt rejection, water flux and resistance to biological attacks, apart from being able to operate at wider range of pH values and temperatures [171]. Boron, as a unique element, has different behaviors according to different pH values. Having a membrane with a better tolerance towards different pH values could aid better in the boron adsorption process without causing fouling issues.

In terms of the size of boron as a very small component in water (~0.098 µm), it is very much convenient and possible for it to be removed using double-layered membranes. For water treatment applications, efforts have been attempted in fabricating membranes with a higher permeability together with a higher selectivity. Most commercial or existing membranes have continuous and interconnected pores which provide the rapid transport of water, but the broad pore size distribution has limited the selectivity when it comes to the precise separation of contaminants from the water. Therefore, the independently functionable layers of bilayer membranes can be tailored chemically by forming a denser active layer followed by a porous substrate layer. Adding pore-forming agents only at the bottom layer during fabrication could be very much helpful in opening water channels without restricting the positioning of smaller boron molecules on narrow-sized pores for further adsorption. By this method, apart from removing the boron through size exclusion, the boron will also have enough time to get chemically adsorbed by adsorbents on the selective layer through methods such as ion exchange.

Furthermore, in terms of operating pressure, it was observed that boron elimination has been increased with increasing feed pressures due to the dilution effect of the permeate water at a higher flux [172]. Therefore, it is very important to have a membrane structure with enough tolerance at higher operating pressures. For these reasons, double-layered membranes are very much better in performing at higher operating pressures due to their outstanding mechanical strength provided by the sub-layer compared to the monolayered membranes. In addition, enhancing the hydrophilicity could help to improve the antifouling behavior of a membrane [173]. Since hydrophilicity is one of the important parameters in ensuring the aligning of layers for co-casting, the antifouling properties of dual-layered membranes will be expected to be very well maintained by the structure provided by the co-casting method.

## 6. Challenges and Future Prospects

With the aim of developing more robust adsorptive membranes for boron removal, progress has been achieved in recent years to overcome the trade-off between the permeability, selectivity, rejection and regeneration issues. Undissociated boric acid molecules in a solution behave similarly to water due to their low molecular weight, the absence of electrostatic charges and its tendency to form hydrogen bonding with the membrane polymeric matrix, thus making the separation process difficult [174]. The extra layer that is formed on the membrane surface prevents the solvent from transferring through the membrane, causing an increase in the pressure across the membrane. This situation lowers the permeate quality. There are various solutions available in order to prevent this undesired adhesion during boron removal, such as the installation of a pre-treatment process, the surface modification of the membrane or chemical and physical cleaning techniques for the membrane. Physical cleaning includes providing backwashing membranes and relaxation for membranes when filtration does not take place [175]. The frequent backwashing of a membrane can cause its structure to get disturbed; therefore, the currently available membranes are often replaced to reach the maximum efficiency. Therefore, surface modification via the introduction of a hydrophilic nature is reported to be able to solve the issue of membrane deterioration [176]. Adsorptive membrane fabrication with a dual-layered approach should be taken into consideration, as hydrophilicity is the main concern that creates such membranes. However, limited studies and research associated with dual-layered membranes for water treatment make it difficult to develop new techniques, as there is not even a benchmark. The complications in controlling the parameters associated with co-casting techniques make it unattractive for people to carry out further investigations, even if the result is quite promising.

However, in the process of seawater desalination, the most common issue faced by the commercial thin-film composite is the chlorine attack which is associated with the low salt-rejecting performance. This is due to the vulnerability of the amide bond with chlorine. Similar kinds of issues are expected in appointing dual-layered membranes for the process of removing boron from seawater, as they have a low tolerance to oxidants and chemicals, and a chemical membrane cleaning process can cause degradation. Surface-tailoring chlorine resistance materials have been accomplished in developing polyamide thin-film membranes for the reverse osmosis process, indicating that this minor issue can possibly be tackled with certain moderations of dual-layered membranes [177]. Several literature reviews suggested solutions regarding the reactivity of chlorine with membrane [178,179,180,181,182]. Various mitigating procedures for chlorine attacks include innovating new membrane materials, surface coating and proper surface grafting, which can prevent the amide bonding. More hybridized materials that comprise polyamide should be discovered. Replacing the polymeric basis of the dual-layered membrane or just simply blending polyamide with other polymers would be a great solution. Dual-layered membranes, which are made from the composites of polyamide-polysulfone, have shown increased water flux [183].

Complications in the sludge removing process that are associated with batch studies on removing boron are the main issue that makes the boron removal via adsorptive membranes rise to the surface. The possibility of regenerating adsorbents in an adsorptive membrane and the ability to separate pollutants from their respective attachment make this process non-polluting. Boron adsorption experiments through batch adsorption have been attempted by various nanoparticles in batch studies [127,184,185]. The inadequacy of nano-sized adsorbents in terms of aggregation, the difficulty of separation and environmental impacts due to leakage into the contact water made the batch studies unable to be further investigated. The behavior of nanomaterials in water and wastewater has still yet to be figured out [186]. Therefore, nanoparticles that are incorporated into adsorptive membranes are discovered, since they are highly efficient in water decontamination, recyclable, cost-effective and compatible with existing infrastructure. However, the problem of leaching an aggregation occurs when adsorbents are inserted into the polymer matrix, which is unavoidable sometimes. In order to solve this problem, the threshold amount of adsorbent to be used must be carefully studied. It is very important to come up with hybridized and very well-pre-treated adsorbents. Several modifications need to be taken into consideration during the fabrication of boron-selective adsorbents. The attachment of several functional groups can be beneficial in order to provide potential sites for boron to be adsorbed. The compatibility of the polymer matrix with the adsorbent to be used should be given more emphasis. Proper functionalization can reduce the agglomeration issue of adsorbents, resulting in the self-assembly of adsorbents on the polymer matrix.

Studies have also shown incredible metal oxide characteristics in removing boron when it is being treated in a batch system. It is highly recommended to incorporate metal oxide to generate highly adsorptive membranes, as they have already been practiced in water treatment applications for the removal of other heavy metals [72,187,188]. The nanomaterial-based adsorptive membranes, mainly with metal oxides, provide advantageous properties such as an enhanced flux, a better pore size distribution, a higher mechanical strength and anti-fouling properties [189]. To mitigate the fouling and wetting issues of membranes, modification using metal oxide has been proven to be effective because it can improve water transport properties and therefore increase the operational performance and long-term stability of the membranes [188,190]. Due to these properties, metal oxide-based membranes are even recommended for desalination, but several metal oxides (e.g., TiO_2_, zeolites and graphene oxides)-incorporated membranes are observed to have limitations related to the fabrication techniques, agglomeration and environmental concerns [191,192]. The upscaling issue of metal oxide-incorporated membranes in real-world scenarios still remains at a challenging issue and has yet to be sorted out completely. So, the other challenge of commercialization is to ensure the stability of these nano-adsorbents over the active layer of membranes in order to avoid the leaching of particles into the downstream. The leaching issue happens due to the limited chemical compatibility of nanoparticles with the membrane structure [193,194]. Therefore, metal oxides alone are incapable of being employed in flow-systems and fixed beds due to the difficulty of the separation of the solids from the water system.

Apart from that, the regeneration issue is also one of the limitations of commercial adsorbents. At the regeneration step, boron-rich brines are produced. The commercially available boron selective resins are designed to remove up to 100 mg/L of boron concentration in order to avoid the frequent regeneration method. As for the clay, fly ash and red mud, the regeneration process is perceived to be uneconomic since they are used once, and the attached boron that is adsorbed can only be separated after they form solid waste. On the other hand, the adsorption property of metal oxides may be reduced due to the dissolving behavior of some species in acid or the base during regeneration.

Most membrane-related studies have yet to evolve from the laboratory scale to industrial applications. Significant progress has been made in recent years in the development of inorganic membranes, leading to polymeric membranes being applied for microfiltration and ultrafiltration applications; however, the unsatisfactory chemical and thermal stability seem to be unavoidable [195]. Therefore, the limitations of fabricated membranes towards boron on a lab scale should be further analyzed in order for them to be utilized industrially. The automation of membrane-casting facilities should be further upgraded by taking various parameters into consideration in order to conduct processes such as co-casting on a big scale.

## 7. Conclusions

The drawbacks of available boron removal technologies have caused the emergence of various treatments which have been experimented with on an academic laboratory scale with the hope of finding a resolution for industrial boron-removing problems. In that case, membrane technology is widely accepted to perform various water-purifying applications in order to a provide higher quality of water. However, the limitations of membranes and their fouling issues in eliminating boron from water still remain questionable. The need to install a secondary treatment to provide boron up to the standard requirement is not only unattractive but very energy-consuming. Therefore, adsorptive membranes, being a single-step efficient treatment, must be explored more in boron-removing applications. Efforts to increase the physicochemical properties and chemical interaction of the membrane framework should be researched more in terms of fabrication and modification to provide long-lasting and efficient boron-rejecting treatments. Various creative approaches such as co-casting techniques should be explored more to come up with promising adsorptive membranes. To date, there has been no industrial implementation of adsorptive membranes due to their lack of improvement and the limited scope being covered in the research field related to boron removal. For future scenarios, advanced research is needed on the recycling of adsorptive membranes for a large number of cycles with excellent adsorbing and desorbing without having to use expensive desorbing chemicals. Additionally, more environmentally friendly adsorptive materials with superior boron rejection, high recovery and an ability to perform at wide range of pH values should be discovered too. The attempts at transforming them into adsorptive membranes should also be analyzed more through different fabrication techniques. Any risks or challenges of transforming adsorbents into adsorptive membranes for boron removal should be tackled extensively in order for them to be used on a large scale. Simplified operation is expected to reduce the complexity and cost of the process. Adsorptive membrane technology is expected to dominate boron-removing applications in the future.

## Figures and Tables

**Figure 1 membranes-12-00798-f001:**
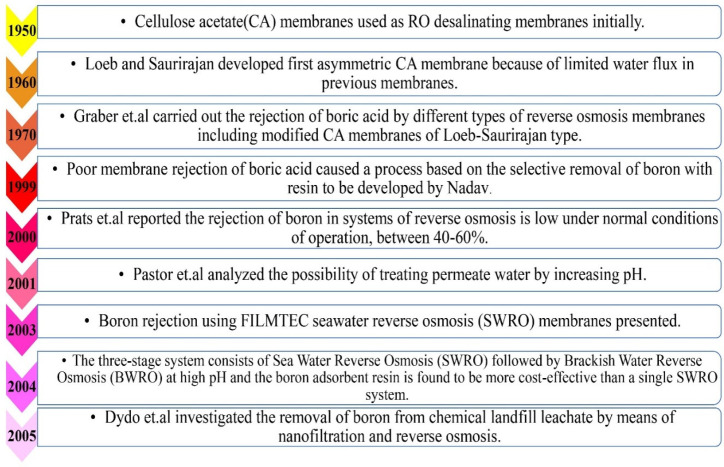
The evolution of post-treatment technology in removing boron [14,15,16,17,18,19,20,21,22].

**Figure 2 membranes-12-00798-f002:**
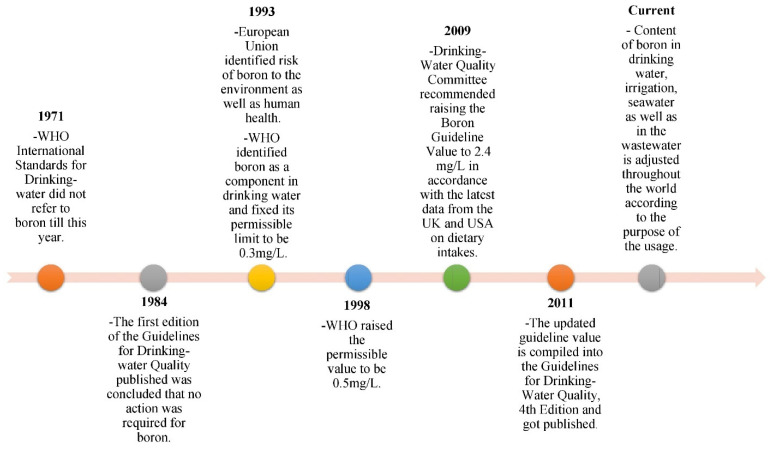
The progress of standards permitted for boron in drinking water [25].

**Figure 3 membranes-12-00798-f003:**
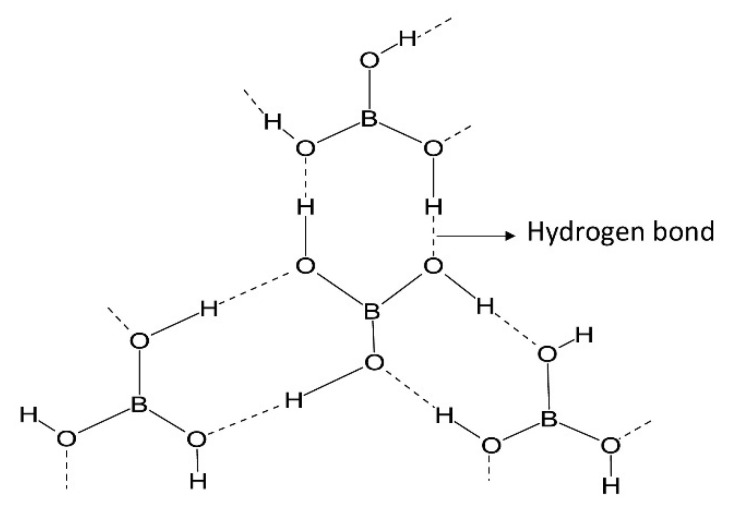
Structure of boric acid with hydrogen bonding.

**Figure 4 membranes-12-00798-f004:**
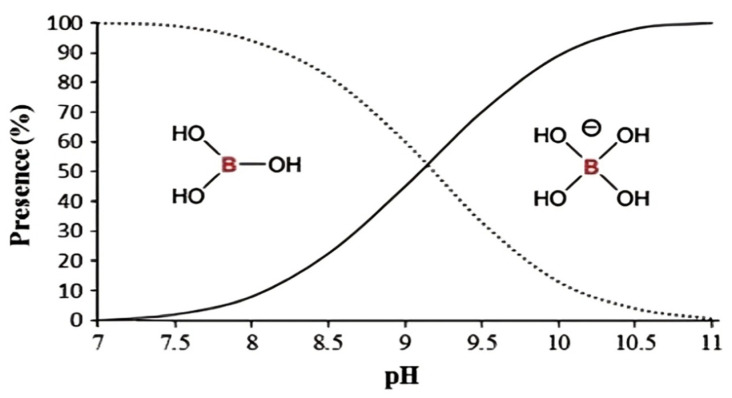
Existence of boron at different pH values. Reprinted with permission from Ref. [32], 2014, Elselvier.

**Figure 5 membranes-12-00798-f005:**
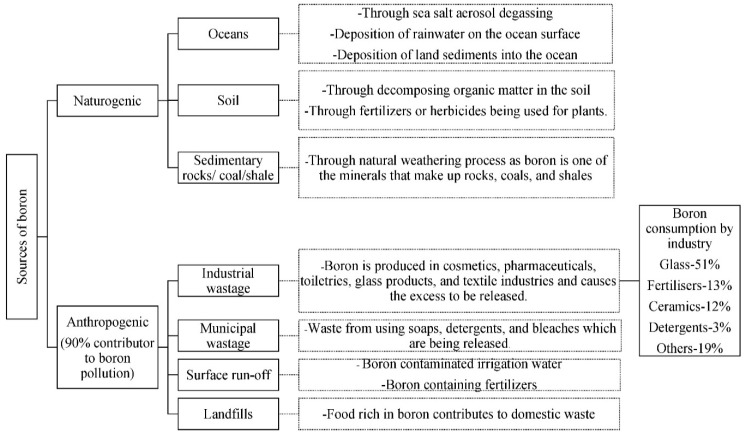
Summary on various sources and their respective processes accounting for boron concentration in the environment.

**Figure 6 membranes-12-00798-f006:**
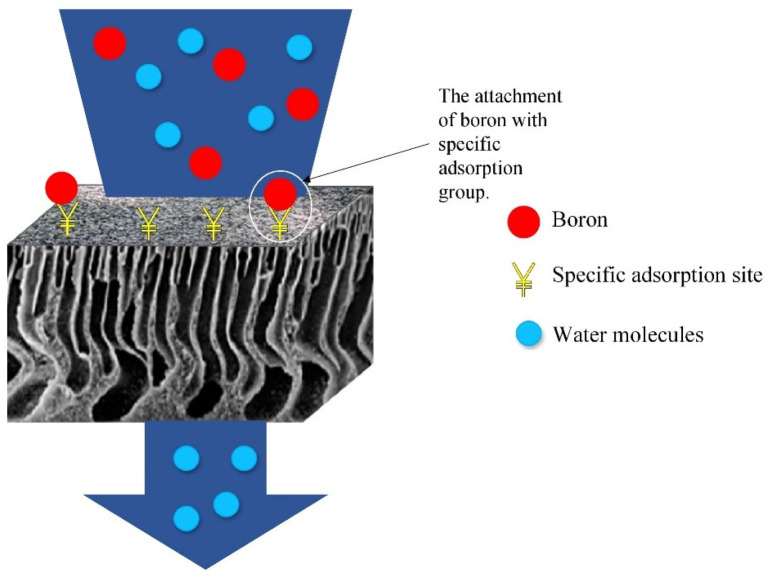
Overall illustration about capturing boron using an adsorptive membrane.

**Figure 7 membranes-12-00798-f007:**
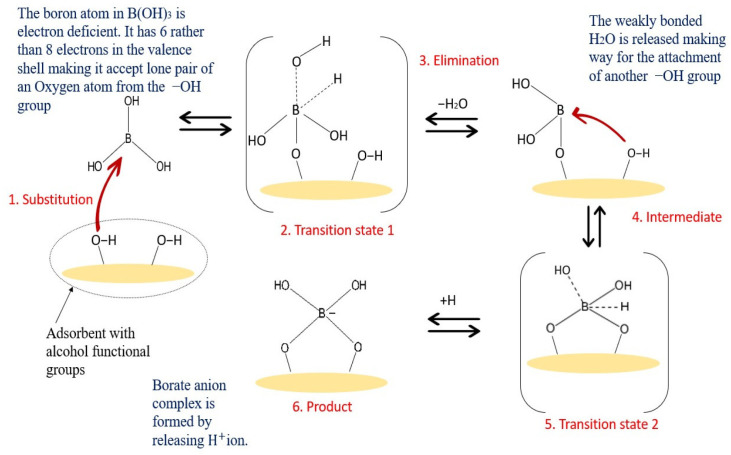
Possible binding mechanism of boric acid and adsorbent with -OH functional groups.

**Figure 8 membranes-12-00798-f008:**
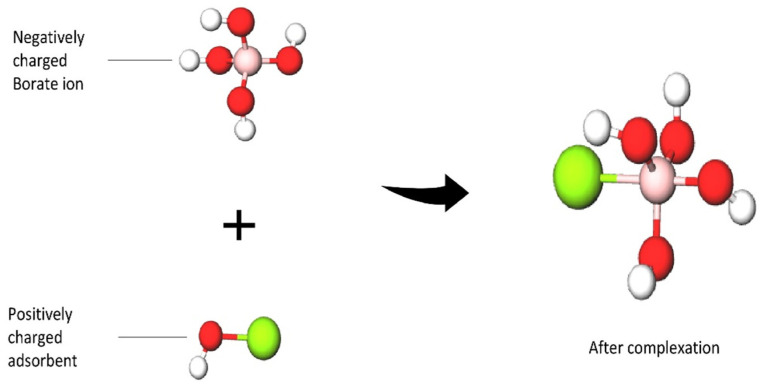
Illustration of the view of the structure after the complexation reaction.

**Figure 9 membranes-12-00798-f009:**
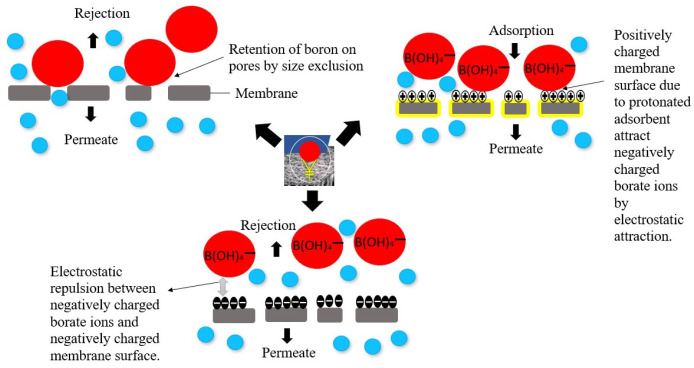
Possible rejection of boron from water through an adsorptive membrane.

**Figure 10 membranes-12-00798-f010:**
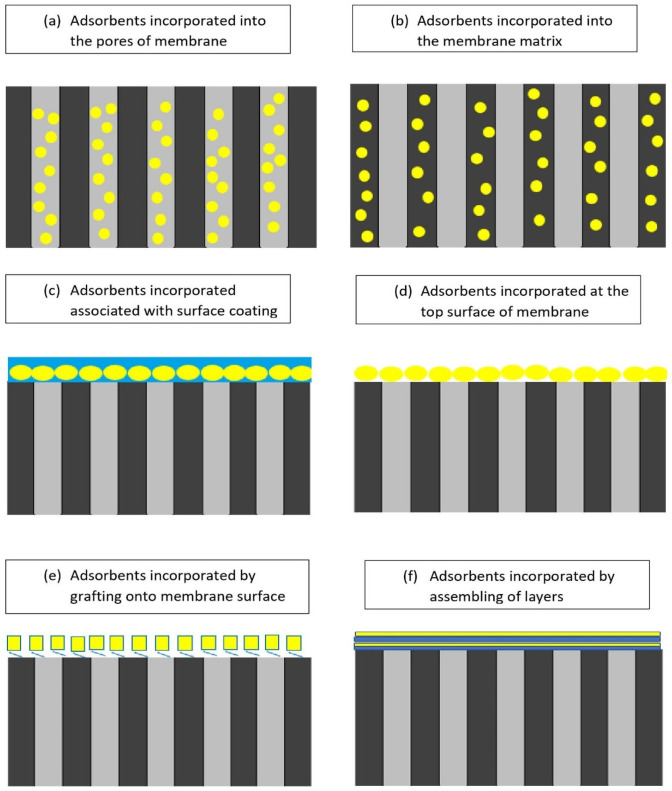
Positioning of adsorbents in the polymer matrix [118]. (**a**) Adsorbents incorporated into the pores of membrane. (**b**) Adsorbents incorporated into the membrane matrix. (**c**) Adsorbents incorporated associated with surface grafting. (**d**) Adsorbents incorporated at the top surface of membrane. (**e**) Adsorbents incorporated by grafting onto membrane surface. (**f**) Adsorbents incorporated by assembling of layers.

**Figure 11 membranes-12-00798-f011:**
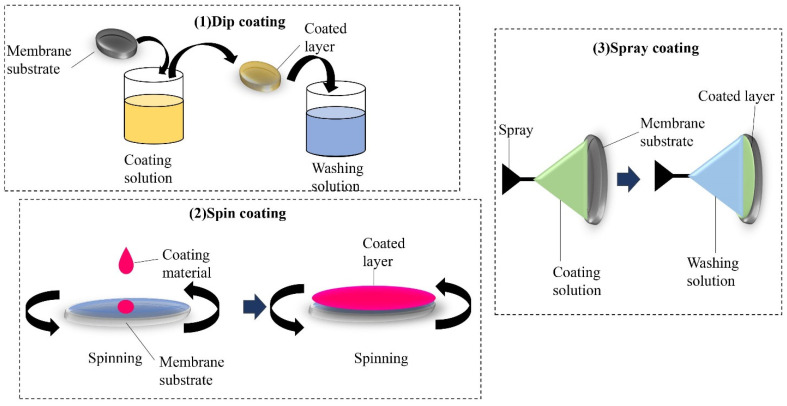
Different coating techniques to form adsorptive membranes.

**Figure 12 membranes-12-00798-f012:**
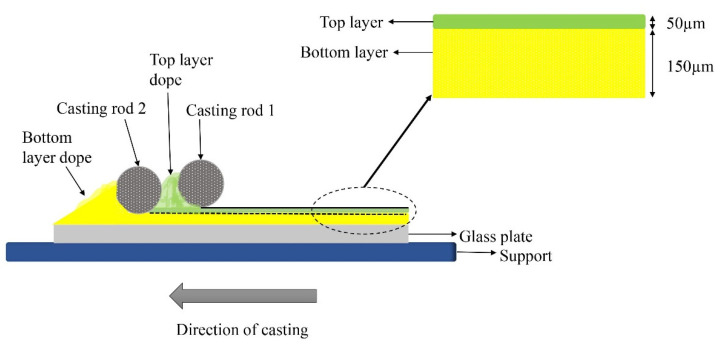
Illustration of the co-casting technique.

**Table 1 membranes-12-00798-t001:** Boron hazard for irrigation water (ppm = part per million). Reprinted with permission from Ref. [27], 1994, Elsevier.

Boron Concentration (1 ppm = 1 mg/L)	Tolerance of Crops
<0.5 ppm	Satisfactory for all crops.
0.5–1.0 ppm	Can show injury for sensitive crops.
1.0–2.0 ppm	Satisfactory among semi-tolerant crops, can cause retarded growth in sensitive crops.
>2.0 ppm	Only satisfactory for certain tolerant crops.

**Table 3 membranes-12-00798-t003:** Preparation of hybrid adsorptive membranes based on previous studies.

Method	Membrane	Removal	Remark	Reference
**1**. **Phase inversion and sintering**	Natural zeolite-based hollow fiber ceramic membrane	Ammonia	Higher removal Good mechanical strength Good permeation flux	[75]
**2**. **Phase inversion**	Polysulfone (PSf)/Organoclay/Organic nanofiller (G, GO, CNTs or CNTOxi) hybrid membranes	Mercury	High selectivity Good mechanical strength Improved thermal stability	[76]
**3**. **Phase inversion and coating**	PSf support matrix membrane loaded with a chitosan functionalized iron nanocomposite membrane fabricated using the phase inversion method and then coated with an alginate active layer	Antimony	High adsorption capacity Enhanced removal via the steric hindrance effect and electrostatic repulsion	[77]
**4**. **Solution casting**	Zeolite nanoparticles-impregnated polysulfone membranes	Lead and nickel cations	Improved adsorption and filtration performances	[78]
**5**. **Sol-gel/electrospinning**	Polyvinylalcohol/tetraethylorthosilicate/aminopropyltriethoxysilane (PVA/TEOS/APTES) nanofiber membrane	Uranium (IV)	Highly reusable and can be extensively used for industrial activities	[79]
**6**. **Interfacial polymerization**	Loose nanofiltration membrane with TiO_2_ nanoparticles on the membrane surface	Salt and dye	High hydrophilicity Good stability of membranes High removal of salt and dye High water flux	[80]
**7**. **Sol-gel method**	Functionalized poly(vinyl alcohol)/tetraethyl orthosilicate (PVA/TEOS) hybrid membranes with 3-mercaptopropyltrimethoxysilane (TMPTMS) groups	Cadmium and nickel ions	High reusability	[81]

**Table 5 membranes-12-00798-t005:** Comparison of single-layered and double-layered adsorptive membranes.

Adsorptive Membrane	Single-Layered	Double-Layered
**Structure**	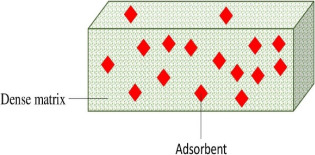	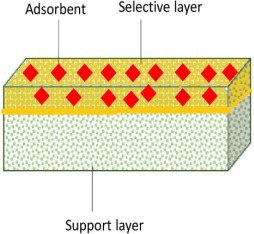
**Advantages**	Increased hydrophilicity	Very well-dispersed adsorbents Materials that make up both layers can be independently selected Reduced cost due to the usage of the adsorbent in a small portion out of a whole membrane High selectivity Good mechanical strength Can be operated at high-range pH values
**Common issues**	Agglomeration and aggregation due to the adsorbent amount being above the threshold amount Poor contact with pollutants, which causes reduced selectivity Low diffusion rate of water due to the packed structure	Delamination issue Low tolerance to chemicals and oxidants

**Table 6 membranes-12-00798-t006:** Methods of fabricating dual-layered flat sheet membranes.

Method	Feature	Material	Application	Output	References
**1. Interfacial polymerization**	Interfacial polymerization is a type of step-growth polymerization in which polymerization occurs at the interface between two immiscible phases (generally, two liquids), resulting in a polymer that is constrained to the interface.	Active layer: Polyamide (PA) Sub-layer: Polysulfone/graphene oxide	Forward osmosis (FO), salt rejection	Enhanced water permeability, higher selectivity, improved performance as a TFC-FO membrane	[153]
**2. Layer-by-layer**	Deposition of thin films and coatings with a precisely controlled composition and thickness (can be used for multilayer films too)	Polyethylenimine (PEI) and trimesoyl chloride (TMC) on a microporous polyethersulfone (PES) substrate.	Nanofiltration (NF)	Increased permeability, stable and higher salt rejection, more compact structure	[154]
**3. Coating and cross-linking**	Process of chemically joining two or more molecules by covalent bonding to be spread on the surface of the membrane.	Polyamide reverse osmosis membrane modified through coating a surface layer of natural polymer sericin	Reverse osmosis (RO)	Increased antifouling ability, decreased pure water permeability, increased salt rejection, capability of decreasing the foulant attachment on the membrane surface	[155]
**4. Co-casting technique**	Simultaneous casting of two dope solutions on a casting plate by controlling several parameters	Silica-impregnated porous bottom layer nano-particle-devoid top surface-interface PA-active layer	Forward osmosis (FO)	Defect-free structure and increased water flux without compromising on the reverse salt flux	[156]

**Table 7 membranes-12-00798-t007:** Studies about dual-layered flat sheet membranes for boron removal.

Membrane Preparation Method	Adsorbent/Active Layer	Polymer	Boron Rejection	Application	Findings	Reference
**1. Interfacial polymerization**	Commercially available NMDG group, (±)-3-amino-1,2 propanediol or serinol	Polyamide-sub layer	90%	Ultrafiltration	40% reduction in salt passage; max boron rejection at pH = 5.2	[158]
**2. Interfacial polymerization**	Trimesoyl chloride	Polysulfone-sub layer	99%	Ultrafiltration	Max rejection at pH = 10	[159]
**3. Interfacial polymerization**	M-phenylenediamine cross-linked 1,3,5-benzenetricarbonyl trichloride followed by a polyamide layer with the UiO66 nanoparticle	Polysulfone-sub layer	91.2%	Reverse osmosis	Improved water flux and salt rejection	[84]

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
