# Peer review of "Adsorptive Membrane for Boron Removal: Challenges and Future Prospects"

_membranes, 2022, doi:10.3390/membranes12080798_

Round 1
Reviewer 1 Report
Dear Editor,
This manuscript is based on a hybrid adsorption/membrane system for boron removal. It is well written, but some changes and suggestions are listed as follows:
1- In the introduction section, it is not common for the authors to use figures. The authors should restructure this section. You could emphasise the importance of novel wastewater treatment systems in the introduction, which could be followed by a literature review, and in the last paragraph you should talk about the research gas and the novelty statement of your study, in addition to a description of what you will show us in this manuscript.
2- When the authors talk about biological treatment, I could not find any biooxidation process in water and water treatment systems. At least you can mention its short definition . You can read the following article:
https://www.sciencedirect.com/science/article/abs/pii/S0048969722041237
3- The effective operating parameters for the adsorption/membrane process should be covered in a separate section.
4- Figures 13, 8, 6 and 5 have low quality.
5- Figure 6 is now well prepared. I suggest replacing it with a table of advantages and disadvantages of each application.
6- Please clarify the effective parameters for the driving force of the separation.
7- Please provide a separate section for adsorption/membrane hybrid prepration methods.
8- The conclusion should be rewritten. Please focus on your key findings.
Reviewer 2 Report
The review by Mehanathan et al. summarized the background information and recent works concerning adsorptive membranes for the removal of boron from water. Some suggestions are listed as follows,
(1) About the structure of the manuscript: Table 2 and the related discussions were included in section 2 with the title of “Adsorptive Membrane Technology”. In section 3 and section 4, single layered and dual layered adsorptive membranes were discussed. I was wondering what the membranes listed in Table 2 should be classified as, single, dual or neither?
(2) It is suggested that the operation mode (dead-end or cross flow) used in the membrane processes should be summarized in Table 2. Also, please include the TMP in Table 2, which is an important factor in membrane processes.
(3) The authors discussed the possible applications of metal adsorbents in section 5. Concerning the use of treated water (irrigation, drinking, etc.), metal leaching is an important issue that should be paid attention to.
(4) The removal efficiencies were shown in Figure 6. However, the removal efficiency may vary significantly under different experimental conditions, and it seemed that the numbers in Figure 6 were only obtained from a few refs., which may be not accurate enough. I would suggest to report more detailed numbers in the figure after reviewing more refs. or remove the current ones directly.
(5) In Table 5, “membrane method” in the header seemed to be inappropriate and may be revised to “membrane preparation method”. In addition, four methods of fabricating dual layered flat sheet membranes were discussed in this section and only one method of “interfacial polymerization” was reported in Table 5. If other reports with other methods could be found, please add them. If not, the above discussions in Table 4 seemed to be cumbersome, and I would suggest to simplify the discussions or add more discussions about the possible reasons why membranes prepared with other methods were not used for boron removal.
(6) Please check the manuscript to revise the typographical errors. For example, subscripts were not correctly marked in Figure 8 and Eq. (1). The brackets were missing in Figure 10.
Round 2
Reviewer 1 Report
All comment have been addresses.